# Effects of a Rice-Based Diet in Korean Adolescents Who Habitually Skip Breakfast: A Randomized, Parallel Group Clinical Trial

**DOI:** 10.3390/nu13030853

**Published:** 2021-03-05

**Authors:** Hyun-Suk Kim, Su-Jin Jung, Eun-Gyung Mun, Myung-Sunny Kim, Soo-Muk Cho, Youn-Soo Cha

**Affiliations:** 1Department of Food Science and Human Nutrition, Jeonbuk National University, 567 Baekje-daero, Deokjin-gu, Jeonju 54896, Korea; khs1120@jbnu.ac.kr (H.-S.K.); egmun1982@gmail.com (E.-G.M.); 2Clinical Trial Center for Functional Foods, Jeonbuk National University Hospital, 20 Geonji-ro, Deokjin-gu, Jeonju 54907, Korea; sjjeong@jbctc.org; 3Research Group of Healthcare, Korea Food Research Institute, Wanju 55365, Korea; truka@kfri.re.kr; 4Department of Food Biotechnology, Korea University of Science and Technology, Daejeon 34113, Korea; 5Department of Agrofood Resources, National Institute of Agricultural Sciences, Rural Development Administration, Wanju 55365, Korea; soomuk@korea.kr; 6Department of Obesity Research Center, Jeonbuk National University, 567 Baekje-daero, Deokjin-gu, Jeonju 54896, Korea

**Keywords:** breakfast skipping, adolescents, rice-based breakfast, body fat mass, cognitive function

## Abstract

During adolescence, healthy eating habits are important, and regular meal intake has an especially positive effect on future health. However, the rate of skipping breakfast has gradually increased. Therefore, this study was conducted to evaluate the positive effects of a rice-based breakfast in Korean adolescents who usually skip breakfast. In this open parallel-group, randomized controlled trial, 105 middle and high school students aged 12–18 years who habitually skipped breakfast were recruited. They were randomly divided into three groups: the rice meal group (RMG, *n* = 35), wheat meal group (WMG, *n* = 35), and general meal group (GMG, *n* = 35). The RMG and WMG received a rice-based breakfast and wheat-based breakfast, respectively, for 12 weeks. After a 12-week intervention, the body fat mass (*p* < 0.05) and body mass index (*p* < 0.05) in the RMG were significantly lower than those in the other two groups, and the stress score was also significantly lower in the RMG (*p* < 0.05). Moreover, after the intervention, in the RMG only, compared to baseline levels, the relative theta (RT) wave activity significantly decreased in eight electrode sites, and the relative alpha (RA) wave activity increased significantly. Eating a rice-based breakfast has positive effects on body fat accumulation and cognitive function in Korean adolescents. Furthermore, a rice-based breakfast plan that is preferred by adolescents should be developed to assist them in developing healthy eating habits.

## 1. Introduction

During adolescence, eating habits change along with the onset of puberty, and eating habits, once formed, are often difficult to modify, therefore, it is important to form healthy eating habits during this period [1]. Further, appropriate eating habits are especially essential for adolescents because regular meal intake has a positive effect on their future health [2]. However, recently, the rate of skipping breakfast has gradually increased due to the “lack of time” and “lack of appetite” among adolescents [3]. The rate of breakfast skipping in 12- to 18-year-old students in Korea was 34.6% in 2016 [4], and more than half of these students did not form a regular breakfast intake habit [5]. 

In several studies, it was found that eating breakfast during youth is associated with a lower body mass index (BMI), weight loss, and positive effects on cognitive performance [6,7,8,9]. Contrarily, it was reported that skipping breakfast causes difficulties in controlling blood sugar, promotes appetite, and increases the risk of obesity and diabetes [10,11]. Although the importance of eating breakfast has been highlighted across several cross-sectional studies [12,13], there are not many studies on dietary interventions.

Recently, the preference for the traditional Korean diet pattern that is based on rice has declined among adolescents while the preference for diet patterns based on noodles, bread, and meat has significantly increased [14]. Although the positive effects of Korean food have been reported in several preceding studies [15,16,17], there are few intervention-based studies in which a Korean breakfast has been recommended for adolescents. Moreover, there are few studies on dietary interventions in which Korean food has been recommended for youths who skip breakfast. It is also important to compare the effects of noodle-, bread-, and meat-based diets that are preferred by adolescents with those of rice-based diets in order to develop a breakfast plan for them in the future.

Therefore, in this randomized controlled trial, we aimed to compare the effects between a Korean-style rice-based breakfast and a wheat-based breakfast provided to youths who habitually skipped breakfast. Furthermore, we aimed to determine the effect of eating breakfast on their physical, emotional, and intellectual well-being.

## 2. Subjects and Methods

### 2.1. Participants

This study was conducted after the study plan was reviewed and approved (CBNU_IRB 2018-11-010-004) by the Institutional Review Board of Jeonbuk National University (Jeonju, Korea). This clinical trial was registered with the Clinical Research Information Service of the Republic of Korea (https://cris.nih.go.kr/cris/en/ (accessed on 3 March 2021) board approval number: KCT0004089), and the dietary intervention was implemented for 12 weeks from April to July 2019. In order to recruit subjects for this study, the study was explained to the students with the consent of the principal of the school that gave permission to proceed with this study among middle and high schools located in Jeonju-si and Wanju-gun, Jeollabuk-do, Korea. Among the students who were willing to participate in this study, only subjects who had breakfast less than three times a week were selected to participate in this study [18,19]. A total of 105 participants who met the selection criteria were chosen based on the results of a screening test in 117 healthy boys and girls (12–18 years old). The exclusion criteria for the study were as follows: (1) subjects who had taken medicines or health functional foods related to adolescent health factors more than five times a week within 1 month before baseline and (2) those who had taken any over-the-counter (OTC) medicine or herbal medicine within 2 weeks prior to baseline or dietary supplements, probiotics, laxatives, dietary fiber, or vitamin preparations within 1 week before baseline. Written informed consent was obtained from all the participants and their parents.

### 2.2. Study Design

This was designed as an open, randomized, parallel-group comparison experiment. (Figure 1). The subjects were assigned at a ratio of 1:1:1 into the following three groups by clinical research coordinators through a random assignment method using a computer-generated random table: the rice-based meal group (RMG; *n* = 35), wheat-based meal group (WMG; *n* = 35), or general meal group (GMG; *n* = 35). The RMG and WMG had a rice-based breakfast and wheat-based breakfast, respectively, on weekdays (except on holidays) for 12 weeks, and the GMG participants maintained their usual eating habits without any dietary recommendations during the study period.

### 2.3. Dietary Intervention

Each diet was developed based on the 2015 Korean dietary reference intake (KDRI, Korean Nutrition Society) and met one-third of the daily recommended intake requirements according to age and sex (Table 1). The diet was based on six food groups (grain, meats, vegetables, fruits, dairy products, and fat) recommended according to sex and age; the total calories in the rice-based breakfast and wheat-based breakfast were the same. The diet was provided as a 4-week cycle menu (Appendix A) in meal box form. Meal compliance was calculated from the total number of meals served and total number of meals eaten. 

### 2.4. Subject Compliance 

During the study period, to minimize the effects of lifestyle changes on the test results, we recommended that the subjects maintain their usual food intake habits for the rest of their meals apart from breakfast. The subjects monitored for lifestyle habits and the suitability of their diet.

### 2.5. Anthropometric Measures and Biochemical Analysis

Anthropometric parameters (weight, height, BMI) were measured using Inbody 720 (BioSPACE Co., Seoul, Korea). Waist circumference (WC) and hip circumference (HC) were measured three times in centimeters to the first decimal place with a tape measure. The waist–hip circumference ratio (WHR) was obtained by dividing WC by HC.

Blood samples were collected from the cubital vein in ethylenediaminetetraacetic acid (EDTA)-coated tubes after a 12 h fast to minimize the effect of circadian rhythm at baseline and endpoint. Blood was centrifuged at 3000 rpm for 20 min (Hanil Science Industrial Co. Ltd., Seoul, Korea) and stored frozen at −80 °C until the analysis. Blood lipid levels were analyzed using an automatic blood analyzer (COBAS NIRA, Roche, Switzerland), and liver function and blood sugar tests were analyzed using a Hitachi 7600–110 analyzer (Hitachi High-Technologies, Japan). The low-density lipoprotein cholesterol (LDL-C) level was calculated according to the Friedewald formula [20]. A homeostatic model assessment for insulin resistance (HOMA-IR) was performed [21].

### 2.6. Perceived Stress and Cognitive Function Test

Stress was measured at baseline and endpoint by adapting the stress tool developed by Cohen, Kamarck, and Mermelstein (1983) to suit the Korean population [22]. This tool consists of a 10-item, 5-point scale, and the score ranges from 0 to 40 points. 

A brief cognitive rating scale (BCRS) survey was conducted to assess cognitive performance at baseline and endpoint. The BCRS score was evaluated on a scale of −3 to 3 in each area (concentration, immediate memory, short-term memory, long-term memory, visuospatial memory, comprehension, language ability, execution ability, endurance, and information processing speed). The total BCRS score was calculated as the sum of the scores assigned to each area [23].

### 2.7. Brain Wave Evaluation

Electroencephalography (EEG), a test for measuring brain waves, was performed at baseline and the end of the study. EEG signals were measured using BIOS24 (BioBrain Inc., Daejeon, Korea), a wired 24-channel polygraph system. According to the international standard 10–20 electrode system, electrodes were placed in eight locations corresponding to the frontal lobe (F3, F4), temporal lobe (T3, T4), parietal lobe (P3, P4), and occipital lobe (O1, O2); reference electrodes were placed behind the ears [24]. 

### 2.8. Investigation of Dietary Intake and Physical Activity

All the participants maintained a 3-day dietary record that included 2 weekdays and 1 weekend day for the evaluation of the nutrient intake at baseline and the study’s end. Dietary intake was analyzed by a registered dietitian using CAN-pro 5.0 software (Korean Nutrition Society, Seoul, Korea). A survey of their physical activity was conducted using the Global Physical Activity Questionnaire (GPAQ) at baseline and the last visit [25]. 

### 2.9. Sample Size and Statistical Analysis

Sample size was calculated using the mean and standard deviation (SD) of body fat loss measured in a previous study’s intervention and control groups [19]. This was assumed to be required for a clinically relevant difference among the three groups at an alpha set at 5% and a power of 80%. This resulted in a required number of 28 subjects in each group and, considering the 20% dropout rate, 105 subjects were enrolled in this study [26].

The data were expressed as mean and standard deviation (SD). As determined by a Shapiro–Wilk test and the values of skewness and kurtosis, a parametric or nonparametric test was performed as appropriate. Baseline differences in demographic and anthropometric measures among the three groups were analyzed with a one-way analysis (ANOVA) or Kruskal–Wallis test. Hochberg or Bonferroni was performed to correct for repeated measures analysis and post hoc *t* test. Spearman’s test was conducted to confirm the correlation between variables.

The primary analysis was performed with an intention-to-treat (ITT) approach, the secondary analysis was performed in the per-protocol set (PP), and the tertiary analysis was performed in the participants with suitable meal compliance (SMC). A two-way (time × group) repeated measure ANOVA with a Greenhouse–Geisser correction was performed to evaluate continuous data in the three groups. The covariables that affected data at the endpoint were age, sex, and school type. Physical activity, based on its known association with data of the anthropometric measurement at the endpoint, was considered as a covariable with sex, age, school type. The changes between the baseline and 12-week data were assessed using an analysis of covariance (ANCOVA). An intragroup analysis was performed using a paired *t*-test or the Wilcoxon test, and a χ2 test was used to determine the frequency difference of categorical variables among the three groups. A value of *p* < 0.05 was set as the statistical significance. All the statistical analyses were performed using SPSS version 18.0 (SPSS, Inc., Chicago, IL, USA).

## 3. Results

### 3.1. Baseline General Characteristics

The baseline characteristics of the participants in the three groups are summarized in Table 2. The mean age of the participants was 15.7 ± 0.2 years, and the ages among the three groups were not significantly different. In addition, differences in the distributions of sex and school type among the groups were not significant. 

### 3.2. Anthropometric Measurements of the Subjects

The baseline anthropometric measurements and clinical and biochemical data were not significantly different among the three groups. Table 3 presents the anthropometric data at baseline and the end of the study as well as the changes in each group. Physical activity and sedentary time were not significantly different among the three groups at the endpoint (not shown). The changes in weight, BMI, body fat mass, and percentage of fat mass were significantly different among the three groups (*p* < 0.05). In particular, the time × group interactions of body fat mass (*p* < 0.05) and BMI (*p* < 0.05) in the SMC participants were found to be significantly different among the three groups (not shown).

### 3.3. The Results of Stress and Cognitive Function

#### 3.3.1. The Stress and BCRS Score

As shown in Table 4, the stress score at baseline was not significantly different among the three groups, but this score in the WMG was significantly higher than that in the other two groups at the end of the intervention period (*p* < 0.05). The total BCRS score significantly increased in the RMG only (*p* < 0.05), and the other groups did not exhibit significant differences. 

#### 3.3.2. The Result of EEG Measurement

The results of the analysis of brainwaves measured using EEG are shown in Figure 2 and Appendix A. The recorded changes in the relative theta (RT) waves exhibited a significant reduction from baseline until the end of the study period in the RMG only. In particular, the time × group interaction at the F3, P3, and P4 electrode sites was significantly different among the three groups (*p* < 0.05). In addition, the effect of the time × group interaction in terms of the relative alpha (RA) waves was significant among the three groups. Regarding the ratio of sensory motor rhythm (SMR) to mid-beta to theta (RSMT), which represents concentration, the recorded changes in the RMG at the four electrode sites, P3, P4, O1, and O2, significantly increased after the intervention (*p* < 0.01, *p* < 0.01, *p* < 0.05, *p* < 0.001). Specifically, the time × group interaction at two electrode sites, P3 and O2, exhibited significant differences among the three groups (*p* < 0.05).

## 4. Discussion

The definition of skipping breakfast is so diverse that it is difficult to establish a precise definition. This is because the concept of skipping breakfast depends on the purpose of the study, the characteristics of the study group, and the characteristics of the questionnaire response [27], as breakfast skipping may be defined by the number of breakfasts consumed, the amount of breakfast intake, or the kinds of food consumed in the preceding studies [18,19]. In this study, skipping breakfast was defined as eating breakfast less than 3 times a week by using the intake number [19].

Adolescents tend to skip meals more frequently and engage in undesirable eating habits because they prioritize the efficient use of their time rather than health considerations [12]. In particular, eating patterns related to breakfast have changed toward the consumption of convenience meals, such as bread and cereals, instead of rice-based meals that are preferred in Korea [14]. Therefore, to reflect this trend, we provided our participants with two types of diets, a diet that teens prefer and one including rice, a staple food in Korea.

A systematic review of several observational studies conducted in Europe until 2010 demonstrated that children and adolescents who eat breakfast have a lower risk of becoming overweight or obese and lower BMIs compared to those who skip breakfast [28]. In addition, in a study on the effect of eating breakfast on BMI and WC in 4430 Japanese factory workers, it was found that skipping breakfast was significantly related to annual changes in the BMI and WC of men [29]. It can be observed that eating breakfast can prevent the excessive weight gain associated with skipping breakfast. The results of the present the study have also demonstrated that the RMG had significantly lower body fat and BMIs than the WMG or GMG. These results suggest that the consumption of rice-based Korean food rather than that of bread- or meat-based meals along with a high preference for eating breakfast can prevent obesity resulting from body fat accumulation in Korean adolescents. As mentioned above, the preferred foods of teenagers are noodles and bread. These foods are known to have high-calorie and high-fat contents but low mineral, vitamin, and fiber contents [30]. It has also been reported that the increased intake of these foods may increase the prevalence of metabolic syndromes, such as obesity and hypertension, in adolescents [31]. In this study, in the WMG that was provided a diet consisting of foods preferred by adolescents for breakfast, the body weight and BMI were higher than those of the GMG. Therefore, it was considered necessary for a study to develop a diet that can increase the rate of Korean food intake among adolescents. 

Breakfast intake may be linked to brain activity during adolescence. However, the results of previous studies related to breakfast intake and cognitive or academic performance are inconsistent because of factors such as the study design, location, type of sampling, choice of objective cognitive tests, and influence of confounding variables [32]. Therefore, this study was designed such that middle and high school students with a similar socioeconomic status could have breakfast in the same place. 

Human thinking and behavior are controlled by cerebral function, but the function of the brain also depends on the activities of several cranial nerves, which are manifested in the form of brain waves [33]. EEG measurements are used to evaluate the psychological state of individuals in response to stimulation by noninvasive and continuous brain wave measurements [34]. In addition, since EEG directly records the activities of the neurons distributed in the cerebral cortex, it is used as an objective indicator of the function of the brain responsible for higher cognition [33]. 

In this study, the RT (4–8/4–50 Hz) wave activity generated when the participants were drowsy was significantly lowered in the RMG, and the RA (8~13/4~50 Hz) activity that indicates a stable and relaxed state was significantly higher in the RMG compared to the GMG. These results suggest that eating a rice-based breakfast could make teens more emotionally stable as well as prevent sleepiness or drowsiness during classes. The RSMT wave activity (12~20/4~8 Hz) is known to increase attention/concentration. This wave activity significantly increased in the RMG after the intervention; therefore, it could be possible to improve attention in students habituated to skipping breakfast through the intake of a Korean breakfast. Moreover, the BCRS scores that were used to evaluate cognitive function exhibited a significant increase in the RMG. Through this study, we were able to confirm that eating a rice-based breakfast has a positive effect on academic performance during adolescence. Chung et al. [23] and Park et al. [35] reported that the cognitive ability of adolescents increased owing to a reduction in mental fatigue and stress resulting from eating mixed-grain rice, because of which the levels of brain-derived neurotrophic factor (BDNF) that is responsible for stress processing and memory activities in the brain increased. In addition, Kim et al. revealed that Korean fermented foods such as kimchi and soy sauce are effective in improving cognition. They considered that fermented foods may improve cognitive function by regulating the release of neurotransmitters such as BDNF, glutamate, gamma-aminobutyric acid (GABA), and serotonin, which are reported to be involved in learning and memory [36]. The exact mechanism related to this has not yet been identified; further research is necessary.

This study has several strengths. First, this study on the effects of breakfast consumption among adolescents was conducted using a randomized intervention design. Second, not only the questionnaire, but also EEG, that is an objective measurement, were applied as cognitive tests. However, there are also certain limitations. First, the participants of this study were middle and high school students, and it was not easy to obtain meaningful results related to nutrient intake (Appendix A) because the food intake records were not specifically and accurately written. In previous studies, it was found that it is difficult to accurately measure food intake via self-records [37]. Second, because other variables apart from breakfast were not controlled, we monitored dietary intake using dietary records. Third, it was also a limitation that rice-based meals are not the preferred food of adolescents; consequently, not only the drop-out ratio was higher, but also the nutrient intake in the RMG group was somewhat lower than that in the WMG group. Fourth, unfortunately, this study did not take into account the potential hierarchies that may exist among the subjects. Therefore, the results were not analyzed by dividing into subgroups. However, through this study, by providing breakfast groups with different types of breakfast, it became an opportunity to confirm the health function indicators of the breakfast intake groups (RMG and WMG) and the breakfast-skipping group.

## 5. Conclusions

This study was a meaningful clinical trial that demonstrates the possibility of preventing body fat accumulation that can cause chronic diseases and of improving academic achievement through Korean breakfast intake in Korean adolescents. Therefore, the results of this study also highlight the need for developing a Korean rice-based breakfast plan that would be accepted by adolescents.

## Figures and Tables

**Figure 1 nutrients-13-00853-f001:**
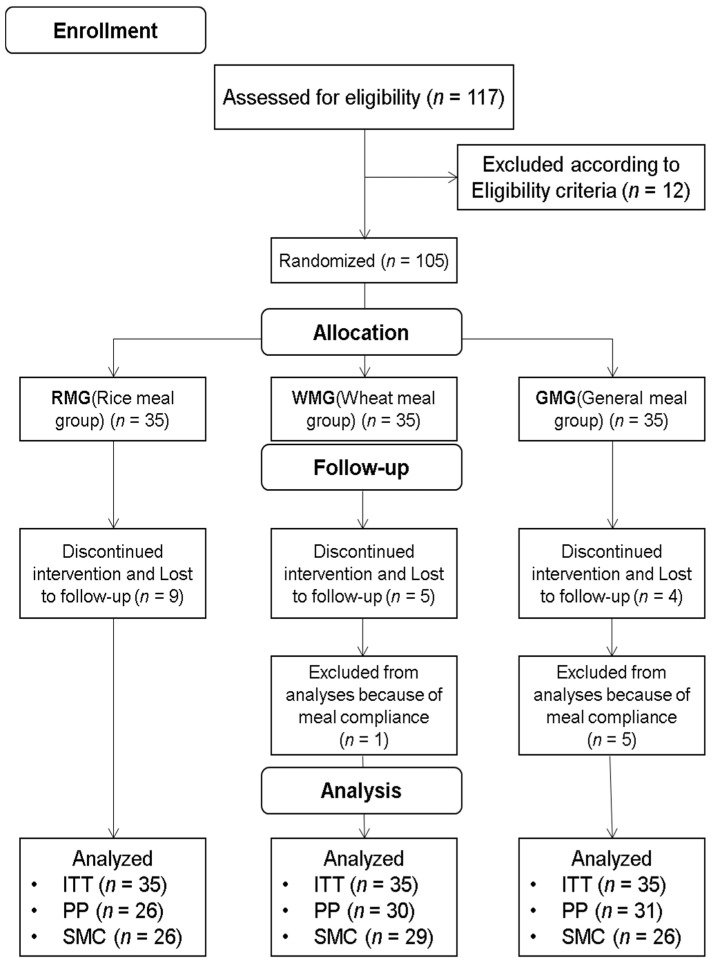
The CONSORT diagram. ITT: intention-to-treat; PP: per-protocol set; SMC: suitable meal compliance.

**Figure 2 nutrients-13-00853-f002:**
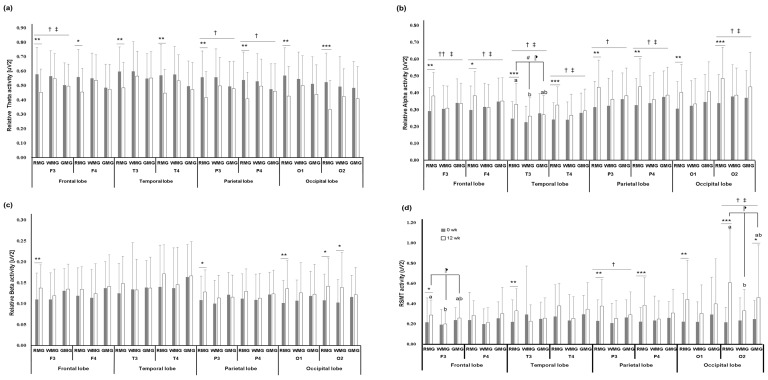
The EEG of (**a**) RT activity; (**b**) RA activity; (**c**) RB activity; (**d**) RSMT activity in 105 study subjects (intention-to-test approach). ^‡^
*p* < 0.05; time X group effect of two-way repeated measures ANOVA, ^†^
*p* < 0.05, ^††^
*p* < 0.01; time X group effect of two-way repeated measures ANCOVA after adjusting for age, sex, and school, ^¶^
*p* < 0.05; analyzed by ANOVA and Hochberg’s post hoc test for comparing mean values of the RMG, WMG, and GMG at 12 weeks, ^#^
*p* < 0.05; analyzed by ANCOVA for comparing mean values of the RMG, WMG, and GMG after adjusting for age, sex, and school at 12 weeks by Bonferroni’s post hoc test for multiple comparison. Values with superscripts (a, b) differ significantly. * *p* < 0.05, ** *p* < 0.01, *** *p* < 0.001; analyzed by paired *t*-test between 0 weeks and 12 weeks, except for significant indications. F3: left frontal lobe; F4: right frontal lobe; GMG: general meal group; RA: relative alpha; RB: relative beta; RMG: rice meal group; RSMT: ratio of (SMR~Mid beta) to theta; RT: relative theta; SMR: sensory motor rhythm; T3: left temporal lobe; T4: right temporal lobe; O1: left occipital lobe; O2: right occipital lobe; P3: left parietal lobe; P4: right parietal lobe; WMG: wheat meal group.

**Table 1 nutrients-13-00853-t001:** The composition of breakfast used in the clinical trial *.

Nutrients	Rice-Based Breakfast	Wheat-Based Breakfast
Energy (kcal)	761.25 ± 3.64	760.94 ± 25.62
Carbohydrates (g)	113.29 ± 8.15	95.80 ± 14.03
Protein (g)	31.92 ± 4.44	26.99 ± 5.16
Fats (g)	19.69 ± 3.59	30.55 ± 7.42
SFA (g)	3.55 ± 1.83	5.39 ± 3.52
MUFA (g)	4.76 ± 2.22	7.35 ± 4.20
PUFA (g)	6.19 ± 3.21	5.68 ± 2.75
Total dietary fiber (g)	10.39 ± 2.29	7.13 ± 2.30
Soluble dietary fiber (g)	1.82 ± 1.13	1.30 ± 1.13
Insoluble dietary fiber (g)	5.22 ± 1.84	3.25 ± 1.46
P (mg)	568.48 ± 108.12	455.79 ± 82.09
Na (mg)	1618.10 ± 486.56	1485.50 ± 323.72
K (mg)	1398.35 ± 476.06	955.66 ± 248.80
Cu (mg)	277.44 ± 120.65	344.97 ± 239.54
Zn (mg)	5.01 ± 1.95	3.19 ± 0.83
Fe (mg)	6.57 ± 2.51	6.95 ± 3.18
Ca (mg)	335.92 ± 100.57	289.05 ± 71.81
Mg (mg)	56.30 ± 28.27	42.71 ± 16.60
Vitamin B1 (mg)	0.73 ± 0.20	0.69 ± 0.22
Vitamin B2 (mg)	0.63 ± 0.18	0.65 ± 0.20
Vitamin B3 (mg)	5.56 ± 1.55	5.88 ± 1.34
Vitamin B6 (mg)	1.24 ± 1.81	0.71 ± 0.35

* Each menu rotated through a four-week cycle, and was developed based on the recommended dietary pattern and the recommended daily intake by age, sex, and food groups according to the dietary reference intake for Korea 2015 (Korean Nutrition Society, Korea). Values are means ± SD. SFA, saturated fatty acid; MUFA, monounsaturated fatty acid; PUFA, polyunsaturated fatty acid.

**Table 2 nutrients-13-00853-t002:** Baseline general characteristics, clinical and biochemical parameters of the study subjects.

Parameters	RMG (*n* = 35)	WMG (*n* = 35)	GMG (*n* = 35)	*p*-Value
Age (years)	15.9 ± 0.3	15.7 ± 0.3	15.5 ± 0.3	0.661 *
Sex, *n* (%)				
Boys	14 (40.0)	15 (42.9)	12 (34.3)	0.823 ^†^
Girls	21 (60.0)	20 (57.1)	23 (65.7)	(χ^2^ = 0.560)
School, *n* (%)				
Middle school	8 (22.9)	9 (25.7)	8 (22.9)	1.000
High school	27 (77.1)	26 (74.3)	27 (77.1)	(χ^2^ = 0.105)
Height (cm)	164.8 ± 7.6	164.6 ± 7.8	165.3 ± 7.4	0.926
Weight (kg)	60.3 ± 10.4	57.2 ± 13.0	57.8 ± 8.2	0.468
WC (cm)	74.4 ± 8.3	71.7 ± 7.5	74.5 ± 7.4	0.261
HC (cm)	93.9 ± 7.1	93.3 ± 7.8	92.9 ± 5.8	0.848
WHR	0.79 ± 0.04	0.78 ± 0.04	0.79 ± 0.03	0.444
BMI (kg/m^2^)	22.2 ± 3.2	21.0 ± 3.6	21.3 ± 2.7	0.299
Physical activity (MET min/week)	2596.9 ± 3886.9	3797.9 ± 8496.8	2191.4 ± 2456.6	0.459
Sedentary time (min/day)	611.3 ± 159.5	628.8 ± 225.4	670.0 ± 206.6	0.451
Glucose (mg/dL)	83.3 ± 6.1	83.9 ± 6.0	81.7 ± 6.3	0.299
HbA1c (%)	5.5 ± 0.2	5.5 ± 0.3	5.5 ± 0.2	0.749
Insulin (μU/mL)	11.7 ± 5.4	10.6 ± 5.3	11.7 ± 7.5	0.701
HOMA-IR	2.4 ± 1.2	2.2 ± 1.2	2.4 ± 1.5	0.800
Triglyceride (mg/dL)	80.5 ± 34.9	70.1 ± 32.5	82.1 ± 32.6	0.095
Total cholesterol (mg/dL)	153.0 ± 18.5	157.5 ± 27.7	161.5 ± 23.0	0.294
HDL cholesterol (mg/dL)	55.1 ± 8.2	58.5 ± 12.1	58.0 ± 12.7	0.263
LDL cholesterol mg/dL)	91.1 ± 20.6	90.3 ± 28.7	97.1 ± 25.4	0.427
AST (IU/L)	23.2 ± 5.4	23.5 ± 5.9	23.2 ± 5.6	0.986
ALT (IU/L)	20.4 ± 14.6	19.2 ± 14.1	17.9 ± 8.9	0.740
γGT (IU/L)	13.6 ± 4.7	13.2 ± 5.1	13.3 ± 5.0	0.928

Data shown as mean ± SD unless otherwise noted. ALT: alanine transaminase activity; AST: aspartate transaminase activity; BMI: body mass index; GGT: gamma-glutamyl transpeptidase activity; GMG: general meal group; HbA1c: glycated hemoglobin; HC: hip circumference; HDL: high-density lipoprotein; HOMA-IR, homeostatic model assessment for insulin resistance; LDL, low-density lipoprotein; MET: metabolic equivalent of task; RMG: rice meal group; WC: waist circumference; WMG: wheat meal group. * ANOVA for comparing baseline characteristics of the subjects among RMG, WMG, and GMG. † Differences between three groups at α = 0.05 by Rao–Scott chi-square test for categorical variables.

**Table 3 nutrients-13-00853-t003:** Anthropometric measurements of the subjects.

Variable	RMG (*n* = 35)				WMG (*n* = 35)				GMG (*n* = 35)				*p*-Value ^2^	*p*-Value ^3^	*p*-Value ^4^	*p*-Value ^5^
	0 Week	12 Week	△	*p*-Value ^1^	0 Week	12 Week	△	*p*-Value ^1^	0 Week	12 Week	△	*p*-Value ^1^				
Weight (kg)	60.3 ± 10.4	61.6 ± 7.9	1.3 ± 6.7 ^a^	0.275	57.2 ± 13.0	59.8 ± 12.1	2.6 ± 5.1 ^b^	0.007	57.8 ± 8.2	59.7 ± 7.7	1.8 ± 4.6 ^a,b^	0.031	0.034	0.643	0.535	0.001
BMI	22.2 ± 3.2	22.5 ± 2.7	0.3 ± 1.6 ^a^	0.301	21.0 ± 3.6	21.9 ± 3.5	0.9 ± 1.4 ^b^	0.001	21.3 ± 2.7	21.7 ± 2.5	0.4 ± 1.3 ^a,b^	0.082	0.023	0.233	0.408	0.0001
Obesity rate, *n* (%)																
under weight	0 (0.0)	0 (0.0)			4 (11.4)	3 (8.6)			1 (2.9)	0 (0.0)						0.411 ^†^
normal	24 (68.6)	25 (71.4)			24 (68.6)	23 (65.7)			25 (71.4)	25 (71.4)						(x^2^ = 8.933)
over weight	7 (20.0)	6 (17.1)			0 (0.0)	2 (5.7)			4 (11.4)	4 (11.4)						
obesity	4 (11.4)	4 (11.4)			7 (20.0)	7 (20.0)			5 (14.3)	6 (17.1)						
Body fat mass (Kg)	15.4 ± 5.9	15.7 ± 5.2	0.3 ± 3.3 ^a^	0.593	15.0 ± 7.6	15.4 ± 6.9	0.9 ± 2.1 ^b^	0.022	15.4 ± 5.8	15.9 ± 5.2	0.50 ± 2.87 ^a,b^	0.333	0.017	0.669	0.873	0.047
Percentage of fat mass (%)	25.9 ± 8.8	25.7 ± 8.0	−0.2 ± 3.4 ^a^	0.693	25.7 ± 9.8	25.8 ± 9.2	0.1 ± 2.4 ^a^	0.806	28.2 ± 8.4	27.9 ± 8.6	−0.27 ± 2.11 ^a,b^	0.453	0.031	0.818	0.457	0.615
Waist circumference (cm)	74.4 ± 8.3	73.9 ± 6.1	−0.49 ± 5.65	0.616	71.7 ± 7.5	72.3 ± 7.3	0.6 ± 3.9	0.410	74.5 ± 7.4	72.9 ± 6.7	−1.7 ± 4.1	0.026	0.467	0.151	0.421	0.254
Hip circumference (cm)	93.9 ± 7.1	91.5 ± 6.0	−2.4 ± 3.9	0.001	93.3 ± 7.8	91.2 ± 6.9	−2.1 ± 3.4	0.002	92.9 ± 5.8	90.3 ± 5.3	−2.6 ± 4.2	0.001	0.079	0.849	0.776	0.0001
Waist:hip ratio	0.79 ± 0.04	0.81 ± 0.04	0.02 ± 0.03	0.002	0.78 ± 0.04	0.80 ± 0.04	0.02 ± 0.03	0.0001	0.79 ± 0.03	0.80 ± 0.04	0.00 ± 0.04	0.499	0.984	0.069	0.610	0.0001
Lean body mass (g)	44.4 ± 8.6	45.5 ± 7.1	1.1 ± 4.9	0.216	43.3 ± 10.1	45.1 ± 9.6	1.8 ± 3.6	0.007	43.7 ± 8.4	44.8 ± 8.4	1.1 ± 2.4	0.011	0.528	0.671	0.925	0.001

Data shown are mean ± SD unless otherwise noted. BMI, body mass index; RMG, rice meal group; WMG, wheat meal group; GMG, general meal group; LBM, lean body mass. △ = 12 week–0 week. ^1^ Analyzed by paired *t*-test between 0 weeks and 12 weeks. ^2^ Analyzed by ANCOVA (difference between change value, RMG vs. WMG vs. GMG) after adjusting for age, sex, school, physical activity by Bonferroni’s post hoc test for multiple comparisons. ^3–5^ Interaction, group effect, and time effect of two-way RM ANOVA adjusted by Bonferroni’s post hoc test. ^†^ Differences between three groups at 12 weeks at α = 0.05 by Rao–Scott chi-square test. Values with superscripts (a, b) differ significantly. All *p*-values of the change values between the three groups without adjusting for age, sex, school, physical activity, and two-way RM ANCOVA after adjusting for age, sex, school, physical activity were not significant.

**Table 4 nutrients-13-00853-t004:** The stress and BCRS scores of the subjects.

RMG			WMG			GMG			*p-*Value ^2^	*p-*Value ^3^	*p-*Value ^4^	*p-*Value ^5^
0 Week	12 Week	*p-*Value ^1^	0 Week	12 Week	*p-*Value ^1^	0 Week	12 Week	*p-*Value ^1^				
Stress score												
Intention-to-treat analysis (*n* = 105) *												
19.71 ± 3.61	19.37 ± 3.14 ^A^	0.865	21.00 ± 3.69	21.02 ± 4.09 ^B^	0.941	19.31 ± 4.79	19.17 ± 3.90 ^AB^	0.872	0.112	0.529	0.016	0.051
Per-protocol analysis (*n* = 87) ^†^												
19.19 ± 3.94	**18.81 ± 4.21** ^a^	0.603	21.03 ± 3.62	21.23 ± 4.39 ^b^	0.813	19.77 ± 4.77	9.10 ± 4.14 ^a^	0.472	0.033	0.960	0.016	0.027
SMC (*n* = 81) ^‡^												
19.20 ± 4.02	19.24 ± 3.67 ^a^	0.949	20.96 ± 3.71	21.36 ± 4.36 ^b^	0.637	19.20 ± 4.81	19.40 ± 3.46 ^a^	0.757	0.047	0.988	0.021	0.044
BCRS score												
Intention-to-treat analysis (*n* = 105) *												
3.57 ± 1.33	7.79 ± 8.10	0.026	3.09 ± 8.55	5.05 ± 8.05	0.233	3.29 ± 9.32	6.33 ± 9.71	0.103	0.166	0.499	0.558	0.427
Per-protocol analysis (*n* = 87) ^†^												
3.00 ± 8.64	7.92 ± 8.89	0.024	2.29 ± 7.91	4.82 ± 8.74	0.151	3.37 ± 8.93	6.33 ± 10.35	0.140	0.202	0.668	0.378	0.763
SMC (*n* = 81) ^‡^												
3.04 ± 10.99	8.31 ± 9.39	0.015	3.41 ± 9.86	4.66 ± 8.63	0.560	3.85 ± 10.65	7.12 ± 11.60	0.142	0.139	0.685	0.373	0.380

Data shown as mean ± SD. GMG: general meal group; BCRS: brief cognitive rating scale; RMG: rice meal group; SMC: study subjects with suitable meal compliance; WMG: wheat meal group. ^1^ Analyzed by paired *t*-test between 0 weeks and 12 weeks. ^2–5^ Analyzed by ANCOVA, respectively, between RMG and WMG, RMG and GMG, WMG and GMG, three groups (RMG vs. WMG vs. GMG) after adjusting for age, sex, and school by Bonferroni’s post hoc test for multiple comparisons. Lowercase letters ^a, b^; the post hoc test of ANCOVA. Capital letters ^A, B^; Hochberg’s post hoc test of ANOVA (*p* < 0.05). All *p*-values between three groups at baseline were not significant. The value in bold: *p* < 0.05 by independent *t*-test between the RMG and the WMG. ^*^
*n* = 35 in the RMG, *n* = 35 in the WMG, *n* = 35 in the GMG. ^†^
*n* = 26 in the RMG, *n* = 30 in the WMG, *n* = 31 in the GMG. ^‡^
*n* = 26 in the RMG, *n* = 29 in the WMG, *n* = 26 in the GMG.

## Data Availability

The data that support the findings of this study are available upon request from the authors.

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
