# Peer review of "Effects of a Rice-Based Diet in Korean Adolescents Who Habitually Skip Breakfast: A Randomized, Parallel Group Clinical Trial"

_nutrients, 2021, doi:10.3390/nu13030853_

Round 1
Reviewer 1 Report
Authors conducted a study to evaluate the positive effects of a rice-based breakfastin Korean adolescents who usually skip breakfast. A total of 105 participants whomet the selection criteria were chosen based on the results of a screening test in 117healthy boys and girls who are between 12-18 years old in Jeonju-si and Wanju-si,Jeollabuk-do, South Korea. The subjects were assigned at a ratio of 1:1:1 into thefollowing three groups by clinical research coordinators through a random assignmentmethod using a computer-generated random table: the rice-based meal group (RMG,35), wheat-based meal group (WMG, 35), or general meal group (GMG, 35). Thenumbers of participants with suitable meal compliance for each group are 26 (RMG),29 (WMG) and 26 (GMG) respectively. All the statistical analyses were performedusing SPSS version 18.0. The manuscript is well-written and well-structured. A fewcomments are listed below.
Comments•
1. Since multiple endpoints were checked in this study, authors need to use multi-plicity adjustment to correctly control the error rate when drawing conclusions.Both rawp-values and adjustedp-values should be provided.•
2. Authors may discuss the potential hierarchical structure or intercorrelationstructure within the data collected. If such structures exsit, the independenceassumption may not hold. Therefore, other statistical methods need to be con-sidered.•
3. Table 3 and 4 are not easy to read, sincep-values appear both at rows andcolumns. Authors should consider a better way to present the results. For example, authors may include the results in multiple tables.

Author Response
To Editor of chief and reviewer 1
Dear sir,
Resubmission of revised manuscript
First of all, we would like to thank you and all the reviewers of our manuscript for going through and making needful comments regarding the manuscript. We made all the changes as suggested by reviewers and tried our best to give appropriate answers for the queries raised by them.
Also, we would like to bring it to your kind attention that the changes made in the manuscript are indicated with color.
If there are any additional things to be addressed, please kindly let me know.
Sincerely,
Youn-Soo Cha

Reviewer 2 Report
Dear Author
This paper outlines a clear research question with an appropriate study design that will add to existing knowledge on nutrient intake and cognitive function of a culturally rice based diet in Korean adolescents who habitually skip breakfast. The methods were clearly described and appropriate statistical analysis was reported. The results and conclusions were well presented and discussed
Yours Sincerely
Author Response
To Editor of chief and reviewer 2
Dear sir,
Thank you very much for taking the valuable time to read this manuscript.
This study started to confirm the positive effects through improvement of breakfast skipping, which is one of the causes of nutritional imbalance in Korean adolescents. Although there are some shortcomings in this study, you considered the purpose of this study worthily. Thank you so much.
Sincerely,
Youn-Soo Cha

Reviewer 3 Report
The study carried out by Kim HS et al. clearly demonstrates that eating a rice meal for breakfast reduced body fat mass and increased cognitive function in Korean adolescents who habitually skip breakfast. This study is a well-designed, valuable interventional trial. The methods used are reliable and the manuscript is generally well-written. The results obtained are socially very important. However, the manuscript has several issues to be addressed.
- The authors should explain how they recruited participants and what kind of adolescents they gathered for selecting 117 participants. These information is important for using the results of this study.
- This Reviewer could not find out the results of GPAQ. Please show them.
- Throughout the manuscript, almost all of the significant digits are nonsense. The authors should take into account the digits of raw data. For example, did they use not only years but also months for age? If not, age is 15.9 +/- 0.3. 165 +/- 1.29 of height is further stupid. They have to change it to 165.X +/- 1.3. Please correct all data.
- Please state the conditions for blood sampling. Were blood samples taken under fasting state? If fasting state, how long fast? If fed state, when?
- The number of participants who discontinued intervention seems to be larger in RMG group. Please explain the reasons if the authors have such data. As they say in Introduction, many Korean adolescents don’t like rice-based meals?
Author Response
To Editor of chief and reviewer 3
Dear sir,
Resubmission of revised manuscript
First of all, we would like to thank you and all the reviewers for going through our manuscript and making the needful comments regarding the manuscript. We have made all the changes as suggested by reviewers and tried our best to give appropriate answers for the queries raised by them.
In addition, we would like to bring it to your kind attention that the changes made in the manuscript are indicated with color.
If there are any additional things to be addressed, please kindly let me know.
Sincerely,
Youn-Soo Cha
